# Recent Progress of Switching Power Management for Triboelectric Nanogenerators

**DOI:** 10.3390/s22041668

**Published:** 2022-02-21

**Authors:** Han Zhou, Guoxu Liu, Jianhua Zeng, Yiming Dai, Weilin Zhou, Chongyong Xiao, Tianrui Dang, Wenbo Yu, Yuanfen Chen, Chi Zhang

**Affiliations:** 1School of Mechanical Engineering, Guangxi University, Nanning 530004, China; 1911301064@st.gxu.edu.cn (H.Z.); 2111301013@st.gxu.edu.cn (Y.D.); 2111301086@st.gxu.edu.cn (W.Z.); 2111301066@st.gxu.edu.cn (C.X.); 2111301014@st.gxu.edu.cn (T.D.); 2111301078@st.gxu.edu.cn (W.Y.); 2CAS Center for Excellence in Nanoscience, Beijing Key Laboratory of Micro-Nano Energy and Sensor, Beijing Institute of Nanoenergy and Nanosystems, Chinese Academy of Sciences, Beijing 101400, China; liuguoxu@binn.cas.cn (G.L.); zengjianhuayu@yeah.net (J.Z.); 3School of Nanoscience and Technology, University of Chinese Academy of Sciences, Beijing 100049, China; 4Center on Nanoenergy Research, School of Physical Science and Technology, Guangxi University, Nanning 530004, China

**Keywords:** triboelectric nanogenerators, power management, self-powered microsystems, switch circuit

## Abstract

Based on the coupling effect of contact electrification and electrostatic induction, the triboelectric nanogenerator (TENG) as an emerging energy technology can effectively harvest mechanical energy from the ambient environment. However, due to its inherent property of large impedance, the TENG shows high voltage, low current and limited output power, which cannot satisfy the stable power supply requirements of conventional electronics. As the interface unit between the TENG and load devices, the power management circuit can perform significant functions of voltage and impedance conversion for efficient energy supply and storage. Here, a review of the recent progress of switching power management for TENGs is introduced. Firstly, the fundamentals of the TENG are briefly introduced. Secondly, according to the switch types, the existing power management methods are summarized and divided into four categories: travel switch, voltage trigger switch, transistor switch of discrete components and integrated circuit switch. The switch structure and power management principle of each type are reviewed in detail. Finally, the advantages and drawbacks of various switching power management circuits for TENGs are systematically summarized, and the challenges and development of further research are prospected.

## 1. Introduction

In recent years, with the rapid development of the Internet of Things (IoT) [1] and 5G technology [2], widely distributed sensor networks [3] and signal transmission electronic equipment have an urgent demand for sustainable energy supply. According to the distributed energy entropy theory [4], traditional centralized power generation may not be able to meet the power supply requirement of the above-mentioned huge number and distributed electronic devices. In addition, the power supply based on a mass of chemical batteries is not sustainable and will cause pollution to the environment [5]. In order to ensure the power supply for distributed sensor networks and communication electronic equipment, researchers have developed many methods to collect energy from sunlight [6], wind [7,8,9], and vibrational mechanical energy [10] based on electromagnetic induction effect [11], photovoltaic effect [12], thermoelectric effect [13] and piezoelectric effect [14], as well as lithiation thermodynamics [15].

The discovery of triboelectrification effect can be traced back to 600 B.C. However, this kind of electricity could not be used until the invention of the triboelectric nanogenerator (TENG) in 2012 by Zhong Lin Wang and his team [16]. When the two materials of the TENG are in contact with each other under external force, the contact surface will generate equal positive and negative electrostatic charges because of triboelectrification. When the two materials are separated by mechanical force, the positive and negative are spatially separated and induce a potential difference between the two electrodes [17]. When the electrodes are short-circuited or connected to a load, the induced potential difference will drive electrons to flow between the two electrodes, which generates energy output. There are four main working modes of TENG: vertical contact-separation mode, sliding mode, single-electrode mode and freestanding-layer-based mode. Originating from the second term of Maxwell’s displacement current, the TENG has been proven to be able to collect various kinds of environmental mechanical energy, including human motion energy [18,19,20], raindrops energy [21,22,23], and ocean wave energy [24,25,26]. It has the advantages of low cost [27], a wide range of material selection [28], flexibility [29] and more. In particular, the TENG can produce higher electromechanical conversion efficiency and higher power energy output than traditional electromagnetic generators under low-frequency, micro-amplitude, and weak-force mechanical excitation [30,31,32]. However, due to its intrinsic capacitance properties, the TENG has a high open-circuit voltage (~kV), large impedance (~MΩ/GΩ), and low short-circuit current (~µA) and average output power (~mW) [33,34,35,36,37,38,39]. In addition, due to the great randomness of the environmental mechanical energy, the amplitude and frequency of the electrical signal output by TENG is unstable. The above-mentioned electrical characteristics cannot match the input requirements of stable DC power for conventional electronic devices. In order to reduce the voltage and impedance, and improve electrical energy supply efficiency, researchers have proposed many methods of switching power management for TENGs [40,41,42,43,44,45,46,47,48,49,50,51,52,53,54,55,56,57,58]. Figure 1 shows the progress of the power management methods from 2013 to 2021.

This review focuses on the latest switching power management for TENGs. First, the circuit model, the output characteristics under resistive/capacitive loads, and the *V*-*Q* curve for the TENG are introduced. Then, based on the switch types, the switching power management methods are divided into mechanical and electronic switches. The design ideas and power management results of the reported works are sorted and illustrated. At the end of the review, we summarize the deficiencies and research challenges of switching power management and give brief prospects for the future development of TENG power management.

## 2. Fundamentals of Switching Power Management for TENG

### 2.1. The V-Q-x Relationship for TENGs

Understanding the essence of TENG can provide theoretical guidance for TENG power management. The fundamental principle of TENG comes from the second term of Maxwell’s displacement current. When using Ohm’s law to derive the characteristics of the circuit under load, it is customary to use the capacitive model, where TENG is equivalent to a series connection circuit of a variable capacitor and an alternating current (AC) voltage source [37].

The *V*-*Q*-*x* relationship of any TENG can be given by Equation (1) [37]:(1)$$V=-\frac{1}{C\left(x\right)}Q+V_{OC}\left(x\right),$$
where *c* is the capacitance between the two electrodes, *Q* is the transferred charges, *V* is the load voltage, and *V_oc_* is the open-circuit voltage. The right side of the equation consists of two voltage terms: the *V_oc_* term is generated by the polarized triboelectric charge, and the *−Q/C(x)* term is the contribution of the transferred charge to *V*.

### 2.2. Output Characteristics under Resistive Load

When the TENG is connected to the resistive load (Figure 2a), the control equation [33] is:(2)$$R\frac{dQ}{dt}=V=-\frac{1}{C_{TENG}}Q+V_{OC},$$

Numerical simulation shows that the current and voltage output by TENG has three working regions as the resistance increases (Figure 2b). In the two regions below 1 kΩ and above 1 GΩ, the output voltage and current barely change with increasing resistance, while in the middle region, as the resistance increases, the current drops and the voltage rises. These output characteristics can be explained by the inherent capacitance of TENG and the impedance matching of resistive load. When the resistance is much smaller than the internal impedance of TENG, the total impedance is determined by the inherent capacitive impedance of TENG; even if the load resistance increases, the output current has no obvious change. When the load is much larger than the inherent capacitance of TENG, almost all the voltage is applied on the load resistance. When the internal and external resistances are approximately the same, the voltage and current vary significantly with load resistance, and the TENG can output the maximum power. The optimal impedance of the contact-separation TENG is given by Equation (3) [35]:(3)$$R_{opt}=\frac{d_{0}^{2}}{F_{opt}^{2}Sv\varepsilon_{0}}\approx \frac{{\left(d_{0}+x_{max}\right)}^{2}}{Sv\varepsilon_{0}},$$
where *d*_0_ is the sum of the thickness to relative permittivity ratios of all the dielectric materials, *x_max_* is the maximum separation distance, *S* is the contact area, *v* is the average separation speed, and *ε*_0_ is the permittivity of vacuum. It can be seen from the formula that the optimal impedance is not related to the surface charge density. Figure 2c shows the dependence of the power on external resistor and speeds.

### 2.3. Charging Characteristics under Capacitive Load

Most electrical switch power management designs have the capacitive load charging circuits (Figure 2d), so it is necessary to understand the charging characteristics of TENG under capacitive load.

The theoretical derivation shows that the TENG with a full bridge rectifier can charge the capacitor under the external periodic movement, which can be equivalent to a DC power supply with internal resistance to charge the capacitor. The voltage of capacitor has a characteristic of saturation (Figure 2e) [39]. The saturation voltage of the capacitor could be calculated by Equation (4):(4)$$V_{sat}=\underset{k\to \infty }{\lim}\left|V_{k,2end}^{C}\right|=\frac{Q_{SC,max}}{C_{min}+C_{max}},$$

The formula shows that *V_sat_* only has a relationship with the maximum value of short-circuit transferred charges, minimum and maximum capacitance.

Most importantly, there exists an optimal load capacitance to achieve the maximum stored energy, as shown in Equation (5) [39]:(5)$$C_{L,opt}=1.592k\left(C_{min}+C_{max}\right),$$
where *k* is the charging cycle and *k* > 10. Calculation results show that the optimal matching capacitance is positively correlated with *k* (Figure 2f).

### 2.4. The V-Q Curve for TENG

When the motion period of TENG is regular, the output energy per cycle can be calculated by Equation (6) [61]:(6)$$E=\bar{P}T=\int_{0}^{T}VIdt=\int_{t=0}^{t=T}VdQ=\oint VdQ,$$

Thus, the energy output per cycle can be expressed by the area circled by voltage and transferred charges in two-dimensional coordinates.

When TENG is connected to the resistor, the *V*-*Q* curve is a closed circle (Figure 2g). This cycle is called ‘cycles for energy output’ (CEO). Due to the resistance of the load, the charge transfer amount *Q_c_* is less than the maximum short-circuit transfer charges amount *Q_sc,max_*. When the travel switches are connected in parallel and short-circuited at the extreme displacement, the area of the *V*-*Q* curve will be expanded (Figure 2h). As shown in Figure 2i, when the load is infinite, the curve enclosing area is close to a trapezoid, which is larger than the area under any impedance, so this cycle is called ‘cycles for maximized energy output’ (CMEO). The energy output per cycle is the enclosing area of the trapezoid and could be calculated by Equation (7) [61]:(7)$$E_{m}=\frac{1}{2}Q_{SC,max}\left(V_{OC,max}+V_{max}^{\prime }\right),$$
where *V_OC,max_* is the maximum open-circuit voltage. *V^′^_max_* is the absolute value of maximum voltage in the condition that *Q* = *Q_max_*.

The *V*-*Q*-*x* relationship for TENG clearly reveals the relationship between output voltage, transferred charges, and displacement. The output characteristics under resistive and capacitive loads show that TENG has an optimal output impedance. The intrinsic capacitance properties of TENG are also clarified. Equation (5) of the optimal storage capacitance guides the research of the switch control strategy in most electronic switching power management circuits. The *V*-*Q* curve illustrates the characteristic of TENG energy output per cycle and determines the theoretical energy limit of a single cycle. It also clarifies the importance of the switch circuit for TENG power management from the theoretical level. Next, the recent research progress on switch power management method for TENG will be introduced and classified according to their structure.

## 3. Power Management with Mechanical Switch

### 3.1. Travel Switch

Travel switch is the main method of TENG power management. The closing and opening of the switch are controlled by the periodic movement of TENG. The switching frequency of the switch is generally once or twice the output frequency of the TENG electric signal. According to the electrical connection method of the switch, the travel switch can be divided into series switch, parallel switch, and switch capacitor converter. Series switch combined with inductor and capacitor circuit can form buck conversion. In addition, it can realize the significant idea of changing the continuous energy release into instantaneous energy release. The parallel switch can redistribute the residual charge through short-circuiting and improve the energy output. The switch capacitor converter can significantly reduce the output voltage of TENG and increase the output charges at the same time.

#### 3.1.1. Series Switch

Cheng et al. [59] introduced a travel switch for the first time, which can convert the continuous output signal into an instantaneous pulse discharge to enhance the instantaneous output power. The reported switch can work for both contact-separation TENG and freestanding sliding TENG. The structure of the generator is shown in Figure 3a. An aluminum needle is integrated on the moving base plate, and the aluminum needle only contacts the electrodes at the two ends of the reciprocating stroke, so that the circuit is connected twice in one cycle. When the switch is turned on, the current waveform (Figure 3b) conforms to the discharge model of resistor-capacitor (RC) circuit. From 500 Ω to 1 GΩ, the power of contact-separation pulse TENG is always higher than that of continuous discharge TENG. It can reach the highest instantaneous pulse power density of 3.6 × 10^5^ W/m^2^ under 500 Ω, which is more than 1100 times of ordinary TENG (Figure 3c).

Qin et al. [69] designed a rectified travel switch. The structure is shown in Figure 3d. The load only contacts the electrode at the end of the reciprocating stroke, which can not only greatly improve the output energy but also convert the AC continuous signal of TENG into a DC pulse signal. The energy storage circuit and the energy transmission process of the rectified travel switch are shown in Figure 3e. When the switch is closed, the TENG energy is first stored in the inductor and then stored in the energy storage capacitor *C*_2_. The energy stored in the inductor is 3.14 μJ. Compared with the single cycle output energy under resistive load (3.33 μJ) (Figure 3f), the inductance loss energy is about 5.7%. The overall energy storage efficiency can eventually reach to 48% (Figure 3g).

The above two works have confirmed that the power management method of energy instantaneous release can effectively extract energy from TENG. However, the output power can be further promoted. Wu et al. [67] reported an output enhancement method based on opposite-charge enhancement and travel switch. The structure, electrical connection and working process of the opposite-charge-enhanced transistor-like TENG (OCT-TENG) are shown in Figure 3h. The two electrodes of the freestanding TENG are respectively covered with two different polymer triboelectric materials: Fluorinated ethylene propylene and polycarbonate (FEP&PC), and the slider is composed of Fluorinated ethylene propylene (FEP) and copper (Cu). According to electron cloud-potential well model (Figure 3i), charge transfer will occur between the same material with different charge density, and a higher number of transferred charges will be produced due to the opposite-charge enhancement. The switch-off current is similar to that of an ordinary freestanding sliding TENG, while the switch-on working principle is similar to the source drain conduction caused by gate trigger of a transistor.

Electrical measurements confirm that the instantaneous power density of the management method exceeds 10 MW/m^2^ (Impedance, 22 Ω to 120 Ω) (Figure 3j). The average power density is as high as 790 mWm^−2^Hz^−1^ and does not change with the external impedance (22 Ω to 10 MΩ), as shown in Figure 3k. This output has set the highest record of TENG output.

#### 3.1.2. Parallel Switch

According to the theoretical analysis of *V*-*Q* curve, Zi et al. [70] confirmed that the single-cycle energy storage efficiency of TENG can be effectively improved (from a maximum of 25% to a maximum of 50%) by setting a switch connected in parallel with TENG and closing it at the two ends of the travel (Figure 4a). The reason is that at the end of each half-cycle, a portion of the charge remains in the electrodes to balance the voltage of the energy storage capacitor and the turn-on voltage drop of two diodes. By applying the above method, the remaining charge can be completely transferred to obtain more transferred charge in the next half of the cycle. The energy cycle curve has two more efficient regions (numbered 2 in Figure 4b) than the case without the switch (numbered 1 in Figure 4b). Experimental results show that the managed energy output is higher than rectified direct charging (Figure 4c) in most energy cycles.

#### 3.1.3. Switch Capacitor Convertor

Based on the principle of switched-capacitor converter, Tang et al. [66] proposed an array of series-parallel-switching capacitors for the step-down and charge enhancement of contact-separation TENG. The working process of the switch is shown in Figure 5a. When the two substrates of the generator are in full contact, the capacitor is switched to the parallel discharge state. In the other working process, the capacitors are charged in series. An electrical characteristic analysis shows that when two capacitors are connected, the output voltage of the TENG will be halved while the output charge will be doubled at the same time. Further research shows that below the megaohm resistance, the output energy under power management hardly varies with the resistance (Figure 5b) and the speed of excitation (Figure 5c). At the same time, the energy loss of the switch is less than 5% (Figure 5d,e).

Zi et al. [71] developed an inductor-free triboelectric management method. Based on the principle of switched-capacitor conversion, this design can increase the charge output by N times while reducing the voltage by N times, where N is the number of capacitors. Different from the work reported by Tang et al. [60], the combined travel switch can not only realize the series-parallel switching of capacitors, but also short-circuiting the two electrodes at the two end points of reciprocating displacement (Figure 5f,g). This design can reduce the residual charge and maximize the charge output per half-cycle. *V*-*Q* curve (Figure 5h) proves that this configuration can collect up to 25% of the energy of CMEO in a single cycle. The power output of the management unit increases by a factor of 19.64 compared to the rectifier (Figure 5i).

Switched-capacitor converters have been proven to effectively reduce the output voltage of TENG and significantly increase the number of transferred charges in a single cycle. However, the number of capacitors cannot be increased indefinitely due to the turn-on voltage drop of the diodes, which limits the ratio of switched-capacitor conversion.

Liu et al. [64] proposed a switched-capacitor conversion circuit based on a fractal design (Figure 5j). The nested structure can effectively reduce the diode’s conduction voltage drop compared with the traditional switched-capacitor converters. Theoretical derivation proves that the total diode turn-on voltage drop of the fractal design based switched-capacitor-convertors (FSCC) is much lower than that of the ordinary switch capacitor convertor.

In a 6-stage 96-unit FSCC circuit, the single-cycle output charge of TENG is 67.8 times higher than the short-circuit transfer charge (Figure 5k). The managed pulse power density is 192 times higher than that of the standard rectifier circuit. By calculating the input (Figure 5l) and output (Figure 5m) voltage and charge, the energy conversion efficiency of FSCC is 94.3%. After adding the filter capacitor (Figure 5n), the circuit converts to constant voltage mode, and the matching resistance of the circuit is reduced from the standard 600 MΩ to 0.8 MΩ. At the same time, 94.5% of the power is retained (Figure 5o).

### 3.2. Voltage-Triggered Switch

The high voltage of TENG can cause the air discharge between electrodes. The electrostatic effect can also deform the micro structures. Therefore, researchers have developed many types of spark switches and electrostatic switches, which can be referred to as voltage-triggered switch.

#### 3.2.1. Spark Switch

Cheng et al. [72] invented an air discharge switch triggered by TENG, and the switch structure is shown in Figure 6a. One side of the switch is a tungsten electrode with a tip diameter of 15 μm, and the other electrode is a stainless-steel plate. With the change of the electrode spacing, there are two kinds of discharge phenomena. If the plasma of the two electrodes is bridged, an arc discharge will occur (Figure 6a); otherwise, it will be a corona discharge (Figure 6b). The relationship between the discharge energy per cycle and the electrode spacing is shown in Figure 6c. The curve has two turning points at a spacing distance of 0.4 mm and 0.72 mm, which can be attributed to the change of the discharge mode. Under a load of 2 MΩ, the peak power and output energy for TENG with air discharge switch is increased by 1600 times and 30 times, respectively, compared with TENG without switches.

Zhang et al. [65] fabricated a self-driven microelectromechanical plasma switch to improve the energy management efficiency of TENG. The circuit architecture is shown in Figure 6d. Firstly, the output energy of TENG is stored in the buffer capacitor (*C_buf_*) through the Bennet circuit. Secondly, when the voltage in *C_buf_* reaches the pull-down voltage of the MEMS switch or the breakdown voltage of the plasma, the hysteresis switch can connect the circuit through direct contact between the anode and cathode or the air breakdown discharge, and the energy is converted by the buck. Finally, the energy is stored in the energy storage capacitor *C_store_*, which is stabilized at DC 3.3 V by a commercial regulator chip.

The author designed and manufactured two kinds of MEMS switches as shown in Figure 6e,f. Figure 6e is a fixed switch with 80 pairs of discharge needle tips. Figure 6f is a movable switch with a pair of discharge electrodes. The author studied and compared the power management performance of the following three switch configurations in detail: a fixed switch with a 7 μm tip gap, a movable switch with a gap of 6 μm, and a movable switch with 9 μm gap. When the energy storage capacitor is connected as the load, the second switch configuration can obtain the highest charging efficiency (56.5 μW/5 Hz@13.5 V) for higher switch-on voltage (Figure 6g). The energy efficiency of MEMS switch and buck conversion is about 35%.

Wang et al. [68] proposed an air discharge-based power management circuit with the circuit structure shown in Figure 6h, where the output energy is temporarily stored in the small capacitor *C_in_* through half-wave rectification. When the voltage of *C_in_* reaches the breakdown voltage of the air switch (about 7.5 kV), the switch discharges. The *V*-*Q* curve shows that the energy output by *C_in_* reaches the maximum of 1.42 mJ when the air gap is 2.4 mm (Figure 6i). The transformer can convert the impedance through electromagnetic conversion. (The efficiency of the transformer is 86.7% which can be calculated by the ratio of load consumption energy to the output energy from *C_in_*) (Figure 6j). In pulse mode, the load is one parallel resistor, and 11.13 kW/m^2^ pulse power (1 Hz, 22 Ω) can be output through power management (Figure 6k). In constant mode, the load is a resistor parallel to a filter capacitor, and the average output power under 200 KΩ load reaches 1.102 mW. Compared with the matching resistance of 35 GΩ before management, 78.5% of the output power is retained (Figure 6l).

#### 3.2.2. Electrostatic Switch

Yang et al. [73] designed an electrostatically actuated vibration switch with the structure shown in Figure 7a. One end of the switch is a copper wire and the other end is a flat plate. Due to the movement of the two triboelectric plates (Cu and PTFE), the positive and negative charges are separated in the horizontal direction, resulting in the potential difference between the two electrodes. This potential difference can drive the vibrator to contact the plate.

The authors connected the above-mentioned switch to the turntable TENG with a quarter grid and adjusted the speed (Figure 7b). A series switch that closes near the peak voltage (open-circuit voltage) extracts the maximum amount of energy from the TENG, so the switch can be managed optimally when the ratio of rotation frequency to switch vibration frequency is 0.25 (Figure 7c).

## 4. Power Management with Electronic Switch

When the working frequency and voltage of TENG are stable, the mechanical switch with a matching voltage and frequency can extract TENG energy with high efficiency. However, mechanical switches suffer from low universality in that they only function at a specific voltage and frequency. The power management circuit based on electronic switch is more adaptable. With a reasonable circuit design, an electronic switch can accurately track the peak of TENG voltage and adapt to the changes of external excitation. This part divides electronic switches into discrete transistor switches and integrated circuit switches by structure and introduces their operating principles and power management effects respectively.

### 4.1. Discrete Transistor Switch

#### 4.1.1. Silicon-Controlled Rectifier

William Harmon et al. [66] designed a power management circuit based on a silicon-controlled rectifier (SCR) (Figure 8a). The energy flow in the circuit can be divided into four stages (Figure 8b). In the first stage, the electrical energy is stored in the capacitor *C_in_* through the rectifier bridge. In the second stage, when the voltage in *C_in_* reaches the reverse bias voltage threshold of the regulator *D*_5_, the current passing through can turn on the SCR, and the energy flows from *C_in_* into the inductor *L*, the output capacitor *C_out_*, and the resistor *R*. In the third stage, after all the energy in *C_in_* is transmitted to the back end, the SCR cuts off, and the inductive energy continues to transfer to *C_out_* and *R*. In the final stage, the energy in *C_out_* is continuously consumed by the resistor *R*.

Simulation results show that the power loss of SCR decreases significantly with increasing inductance, but the increase in inductance will also increase the equivalent series resistance in the circuit. The waveforms of *C_in_* and *C_out_* voltages are shown in Figure 8c,d. The voltage of *C_in_* decreases once per cycle, which means that the energy of *C_in_* is released to the back end once per cycle. The *C_out_* voltage presents a ripple shape, and the ripple value decreases with the increase of capacitance. Due to the appropriate turn-on time and very low energy loss of SCR, this switching circuit can reduce the output matching impedance of contact separation TENG from 150 MΩ to 2 MΩ, while maintaining 84.3% of AC peak power under 150 MΩ load, which is shown in Figure 8e.

Wu et al. [74] proposed a method of power management for TENG using triode as a switch. The circuit is shown in Figure 8f. The peak detection circuit that is composed of *R*_1_, *C*_1_, and voltage comparator *Amp* is used to detect the voltage peak of TENG. The delay circuit that is composed of *R*_2_, *C*_2_, AND gate circuit and inverter circuit is used to accurately adjust the pulse duration of the switch control signal. This work adopts a design idea that is similar to that reported by Cheng et al. [75]. The difference is that the author not only verified the feasibility of using triode as a switch to manage the output power of TENG through simulation and experiment, but also studied the influence of resistance in both differential and delay circuit on power management performance in detail.

As shown in Figure 8g, the greater the resistance of *R*_1_ in the differential circuit, the closer the pulse voltage at point *h* to the voltage peak of TENG. However, the practical tests show that when the resistance increases to more than 600 kΩ, multiple pulses are generated (Figure 8h). Therefore, *R*_1_ is set at 600 kΩ as the optimal value of the differential circuit. *R*_2_ has an important influence on the ON time of the switch in the time-delay circuit. By comparing the energy optimization in the end energy storage capacitor *C* after charging for 10 s, the authors obtained the optimal value of *R*_2_ (Figure 8i). Through the above optimization, the authors obtained a power management efficiency of 37.8%.

#### 4.1.2. MOSFET

Niu et al. [62] first introduced a triboelectric power management method of two-stage electric energy release (Figure 9a). According to the theoretical derivation, the average power can reach the peak when the charged capacitor voltage reaches 0.7153 times the saturation voltage. Therefore, the author designed two switches controlled by logic circuit to extract the energy in the buffer capacitor periodically. Whenever the voltage in the buffer capacitor reaches the above optimal value, an electronic switch is closed to transmit the energy to the rear storage capacitor through electromagnetic conversion, which can effectively improve the energy utilization efficiency.

The average power output at the optimal voltage is 75% of the AC output under TENG matching impedance by theoretical calculation. The efficiency of Ctemp to Cstore transfer is 90%. The energy loss can be attributed to the resistance of the inductor and the leakage current of the switch control loop. The ratio of the maximum DC power output after power management to the maximum AC power output under resistive load is 59.8% (Figure 9b,c), and the matching impedance is reduced from 4.26 MΩ to 180 kΩ.

Xi et al. [63] invented a power management circuit combining TENG maximized energy output cycle (CMEO) and buck convertor. The circuit architecture is shown in Figure 9d. The series switch in the loop, the inductor, and capacitor at the back end form a standard buck circuit. Figure 9e shows the structure of the switch control loop. The comparator is used to detect the TENG output voltage, and when the TENG voltage is greater than the reference voltage, the control signal turns on the MOSFET. Significantly, the power for the logic circuit comes from the TENG, without any other external resource.

The innovative point of this design is that the switch is only closed near the TENG voltage peak. According to the *V*-*Q* curve of TENG, the quadrilateral area composed of the open-circuit voltage and the short-circuit transferred charges represents the maximum energy that the TENG can output per cycle (Dotted line in Figure 9f). The series switch controlled by the TENG voltage can achieve the function of CMEO (Solid line in Figure 9f). The area ratio of the two quadrilaterals (Solid line area to dotted line area) is 84.6%. After connecting the resistive load, the management circuit can effectively reduce the output matching impedance of TENG from 35 MΩ to 1 MΩ, while retaining the efficiency of 80.4%.

Song et al. [76] proposed a self-charging smart bracelet based on the integration of freestanding sliding TENG and flexible PCB-based power management module. The series switch in the circuit is closed only when the open-circuit voltage of the TENG reaches the peak value (Figure 9h). According to CMEO theory, this method can extract TENG output energy with maximum efficiency. The structure of the control circuit is shown in Figure 9i. The rectified signal is differentiated first and then input to the non-inverting terminal of the comparator. The differential signal is compared with the zero potential of the inverting input terminal. This above method can accurately detect the voltage peak. The switch control signal output by the comparator is connected to the gate of MOSFET switch after being processed by the delay module composed of RC delay circuit and AND gate circuit. The function of the delay module is to accurately control the closing time of the switch to achieve the maximum energy transfer efficiency of the LC oscillation circuit.

As shown in the output power-impedance diagram (Figure 9j), the DC power of the energy collection circuit with power management under 10 kΩ and 47 uF load can reach 69.3% of the AC maximum power output.

Based on the idea of “maximum power point tracking” (MPPT) in solar and piezoelectric power management circuits, Sontyana Adonijah Graham et al. [77] designed a TENG power management circuit for charging lithium-ion batteries. The circuit architecture is shown in Figure 9k. Output energy is temporarily stored in the capacitor *C_IN_* through rectification, and the back end of *C_IN_* is a standard buck conversion circuit. In particular, the MOSFET control signal is output by an electronic MPPT controller, which can detect the *C_IN_* voltage in real time and maintain *V_IN_* at half of the generator’s open-circuit voltage. The author follows the idea of tracking the maximum power point in piezoelectric and believes that when the output voltage is controlled to half of the open-circuit voltage, energy can be harvested with the highest efficiency. The voltage waveforms are shown in Figure 9l. The blue line shows that *V_IN_* oscillates around 100 V. Each pulse peak of *Φ_N1_* (pink line) corresponds to a turn-on of the MOSFET, and it also corresponds to a steep drop in *V_IN_*, which means energy is transferred once.

Madhav Pathak et al. [78] deduced the characteristics of charging battery using Full Wave Rectifier (FWR) circuit, Parallel Synchronous switched Harvesting on Inductor (P-SSHI) circuit and Series Synchronous switched Harvesting on Inductor (S-SSHI) circuit. (Figure 9m). The parameters studied include the following three: the energy output per cycle, the optimal battery voltage, and the maximum voltage that the battery can reach. Comparative experiment shows that whether the inductor is connected in parallel or in series in the loop, the energy storage effect of the battery is better than the FWR (100 times gain). The optimal load voltage of the S-SSHI circuit is not affected by the quality factor of the inductor and is smaller than that of the P-SSHI circuit. The author built a power management circuit to verify the theoretical derivation and simulation (Figure 9n). The generation principle of MOSFET control signal is the same as that in Song’s work [76]. The difference is that the author designed two independent contact-separation TENGs in order to avoid the mutual interference between energy harvesting and signal generation.

### 4.2. Integrated Circuit

S. Boisseau et al. [79] reported a self-starting power management integrated circuit (IC) for harvesting piezoelectric and triboelectric energy with the circuit structure shown in Figure 10a. The startup circuit is composed of an energy harvester, a rectifier bridge, a depletion MOSFET (*dMOS*) *K_bp_*, and a capacitor *C_s_*. When the energy in *C_s_* is not enough to power the control circuit, the *K_bp_* is turned on so that the current bypasses the flyback circuit and flows directly into *C_s_*. When the energy in *C_s_* can power the control circuit, the two switches *K_p_* and *K_s_* in the flyback circuit are closed at the voltage near the generator voltage peak to extract the generator voltage with maximum efficiency. The voltages of C_b_ and *C_s_* in the start-up phase are shown in Figure 10b. The voltage waveform proves that the control circuit works intermittently in the start-up phase until the rising slopes of the two capacitor voltages are the same, and the start-up process is completed. The integrated circuit is manufactured in an AMS 350 nm complementary metal–oxide–semiconductor (CMOS) process. The off-chip components include a flyback circuit, a rectifier bridge, two buffer capacitors, derivative capacitors, and *dMOS*. The power consumption of the peak detection circuit is 150 nW@3 V.

Inho Park et al. [80] designed a high-voltage dual-input integrated circuit converter for power management of TENG. The circuit structure is shown in Figure 10c. The positive half-wave and negative half-wave energy from TENG are stored in the capacitors *C_in,P_* and *C_in,M_*, respectively, by the dual-output rectifier. Through reasonable control circuit design, the above two capacitors are accurately controlled to release energy to the buck circuit when the maximum output power is reached. This method could extract TENG energy with the highest efficiency. Based on the maximum power point tracking analysis of TENG and the fractional open-circuit voltage method (FOCV), the authors experimentally obtained the ratio of the output voltage to the open-circuit voltage at the maximum average output power of TENG. The above-mentioned integrated circuit adopts the 180 nm Bipolar-CMOS-DMOS (BCD) process with an effective area of 2.482 mm^2^. The off-chip components include an inductor, five input capacitors, one output capacitor, and four resistors. The total power consumption is 754.6 nW. After connecting the TENG and the load, the author measured the accuracy and efficiency of the MPPT. The accuracy is higher than 96.39%, and the MPPT efficiency reaches the highest of 94.86% when the input is 17.13 μW. When the input power is 20.9 μW, the overall end-to-end efficiency is 52.9%. This work proves the feasibility and effectiveness of the maximum power point tracking method for triboelectric energy harvesting.

Ismail Kara et al. [81] designed a triboelectric energy harvesting circuit based on a synchronous inductor parallel switch and DC step-down conversion, which can convert 70 V input into 2 V DC output. The overall topology of the circuit is shown in Figure 10d. The circuit is composed of the following three step-down modules: synchronous inductance parallel switching rectifier circuit, DC step-down conversion circuit, and switched capacitor conversion circuit. When the voltage of capacitor *C_rect_* reaches 70 V, the step-down conversion is started, the voltage of *C_out_* is stabilized at 10 V, and the load voltage is stabilized at 2 V. Similar to the idea of piezoelectric management, when the switch is closed at the time of zero-current the voltage can be reversed instantaneously, so as to avoid energy loss.

The voltage ripple of 5 V is 180 mV, and the ripple of the output is 24 mV. The output can drive a wireless sensor chip to work for 4 ms, and the output power is 2888 nJ. The efficiency of the above three buck modules is 69.2%, 67.5% and 70%, respectively, and the overall end-to-end efficiency is 32.71%.

## 5. Conclusions and Prospects

In this review, we systematically reviewed the recent progress of switching power management for TENG. Based on the type of the switch, the current power management strategies are classified into mechanical switches and electronic switches. The mechanical switches mainly consist of the travel switches and voltage triggered switches. The electronic switches can be classified into discrete transistor switches and integrated circuits. The switch can effectively improve the output efficiency and function as an impedance converter, which can satisfy the power demand of conventional electronic devices.

For the power management efficiency, at a specific frequency, the series mechanical switch can be closed at twice (or once) the TENG signal frequency. This configuration can achieve the efficiency enhancement that energy is first accumulated and then released. According to the CMEO theory, the switch that is closed at the voltage peak can extract the TENG energy with the maximum efficiency. The parallel mechanical switch can redistribute the charge in the two electrodes by short-circuiting to avoid the negative impact of residual charge on charging efficiency. Therefore, the mechanical switch can achieve high power extraction efficiency.

For the universality of TENG power management, there are two main strategies for the electronic switch. Based on MPPT theory and optimal capacitance theory, the output energy from TENG is first stored in a front-stage capacitor (nano-farad level). On the one hand, it is conducive to the collection of mechanical energy from an irregular and unstable natural environment. On the other hand, the strategy of energy hierarchical collection has the function of impedance matching. The switching designed based on the above theory mainly includes SCR that conducts under fixed voltage and MOSFETs controlled by logic signals. Based on CMEO theory, the output energy from TENG is directly transferred into inductance or transformer to converted into magnetic energy. The switches designed based on the above theory are MOSFETs controlled by a logic circuit signal. The electronic switches circuit designed by any power management strategy requires signal detection and control signal generation circuit, which can accurately identify the voltage peak or the output power peak of TENG. In addition, considering the energy transmission time in the LC oscillation process at the moment of switch closing, the closing time of the switch can also be accurately controlled to improve the energy transmission efficiency. Therefore, in terms of universality, the electronic switch has incomparable advantages over the mechanical switch.

However, further research still faces the following problems:

Firstly, the structure of mechanical travel switch is rigid and complex. The metal contacts are not conducive to the integration with circuits. For the voltage-triggered switches, a custom switch can only operate at a specific frequency or voltage amplitude. Therefore, it is necessary to convert the unstable and random kinetic energy in the environment into stable electrical output through novel and durable mechanical design.

Secondly, the operation of a control circuit in an electronic switch mostly requires an external power supply. Therefore, it is urgent to design a low-power-consuming and self-starting circuit for power management to realize self-driving in the real sense. In addition, semiconductor-based electronic switches cannot achieve absolute isolation in an open circuit state, so reducing quiescent current and power consumption is the key method to improve power management efficiency. Meanwhile, due to the intrinsic high-voltage property of TENG, it is easy to cause electrostatic breakdown damage of a transistor circuit. Therefore, the high-voltage protection is the guarantee of efficient and stable operation of a power management circuit. Last but not least, the electronic switch power management strategies mainly have the origin of piezoelectric or solar energy collection strategy. However, due to the essential electrical property differences among the TENG and piezoelectric sheet and solar cells, it is necessary to develop an optimal switching control strategy suitable for the TENG.

Thirdly, based on the above analysis, it can be concluded that mechanical switches can achieve high energy management efficiency and electronic switches have strong universality. Therefore, researchers need to balance the contradiction between efficiency and universality. For example, mechanical switches can be used in the power management circuit for harvesting stable vibration energy in industrial equipment in order to achieve high efficiency, and electronic switches can be used in circuit for harvesting mechanical energy with unstable motion such as human body movements, wind energy and more.

Finally, as an important direction of TENG, the research on marine energy, namely blue energy, has broad development prospects. However, most of the current power management strategies are voltage step-down strategies for small electronic devices, and there is still a lack of effective solutions for high-voltage power generation. Therefore, the development of a power management circuit based on blue energy harvesting will be the key to the large-scale application of TENG energy supply technology in the future. In order to realize the transformation from micro energy to large energy, from weak electricity to strong electricity, researchers need to solve problems such as power generation efficiency, energy storage, transformation, transmission, environment protection, and topology optimization of large-scale power generation.

As a significant technology in the development of TENG, the power management circuit can effectively improve the output power and efficiency of TENG. We believe that in the near future, the efficient and versatile power management circuits can be developed for TENGs and used in most energy harvesting occasions, which become a key technology to promote the development for IoT and 5G communication.

## Figures and Tables

**Figure 1 sensors-22-01668-f001:**
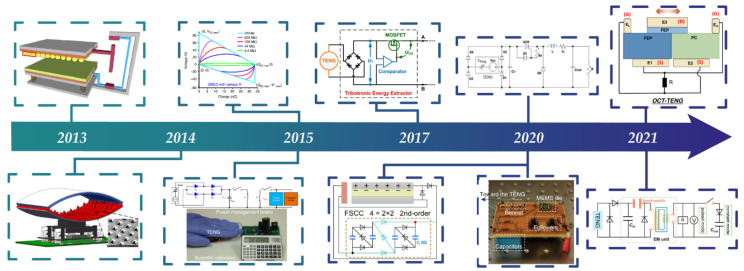
Brief timeline of the switching power management methods for triboelectric nanogenerator (TENG). Reproduced with permission [59]. Copyright 2013, American Chemical Society. Reproduced with permission [60]. Copyright 2015, *IOP science*. Reproduced with permission [61]. Copyright 2015, *Nature*. Reproduced with permission [62]. Copyright 2015, *Nature*. Reproduced with permission [63]. Copyright 2017, Elsevier. Reproduced with permission [64]. Copyright 2020, *Nature*. Reproduced with permission [65]. Copyright 2020, *Nature*. Reproduced with permission [66]. Copyright 2020, Elsevier. Reproduced with permission [67]. Copyright 2021, *Nature*. Reproduced with permission [68]. Copyright 2021, Elsevier.

**Figure 2 sensors-22-01668-f002:**
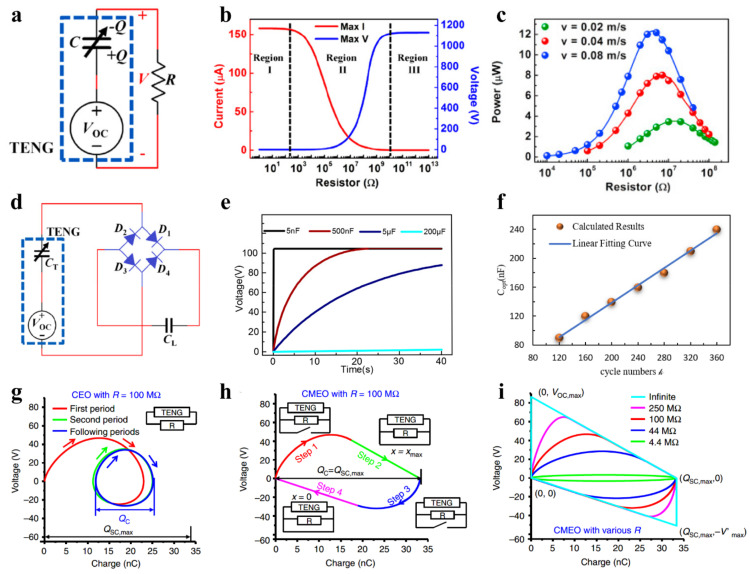
Fundamentals of switching power management for TENG. (**a**) TENG connected with resistances load. (**b**) ‘Three-region’ characteristics under resistive load. (**c**) Dependence of the power on external load and velocity. (**d**) Full-wave rectifier circuit. Reproduced with permission [33]. Copyright 2014, Elsevier. (**e**) Saturation curve of capacitor voltage. (**f**) Relationship between optimal matching capacitance and numbers of charging cycle [39]. (**g**) Cycles for energy output under 100 MΩ. (**h**) Four steps of cycles for maximized energy output (CMEO) under 100 MΩ. (**i**) CMEO under infinite load. Reproduced with permission [61]. Copyright 2015, *Nature*.

**Figure 3 sensors-22-01668-f003:**
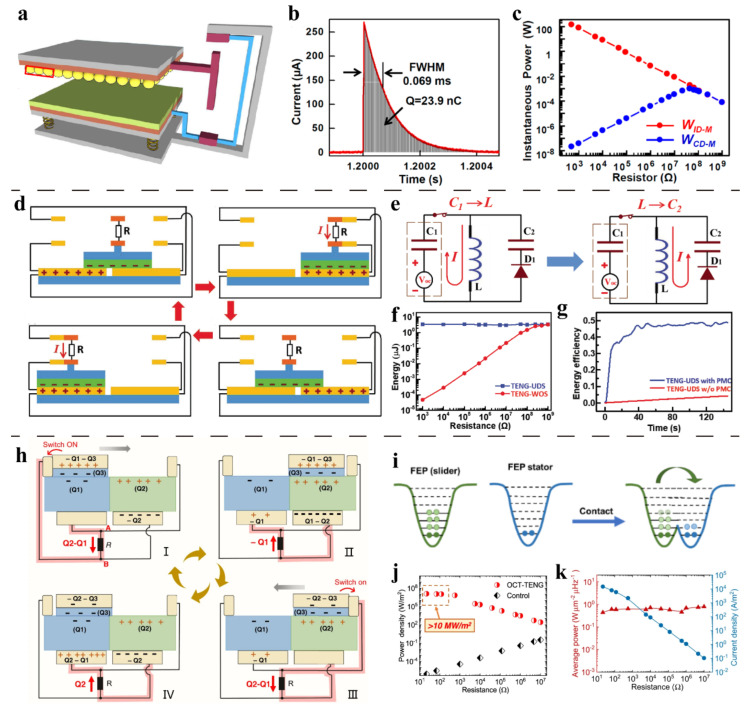
Power management with series travel switch. (**a**) Structure of the travel switch. (**b**) Pulse current at the switch-on moment. (**c**) Dependence of instantaneous power on resistor with/without travel switch. Reproduced with permission [59]. Copyright 2013, American Chemical Society. (**d**) Structure and working process of the rectified travel switch. (**e**) The energy storage circuit and energy transmission process. (**f**) Dependence of the output energy on the resistance. (**g**) The overall energy storage efficiency with/without travel switch. Reproduced with permission [69]. Copyright 2018, John Wiley & Sons. (**h**) The structure, electrical connection and working process of the opposite-charge-enhanced transistor-like TENG (OCT-TENG). (**i**) Electron cloud-potential well model. (**j**) Pulsed power comparison. (**k**) The dependence of average power and current density on resistance. Reproduced with permission [67]. Copyright 2021, *Nature*.

**Figure 4 sensors-22-01668-f004:**
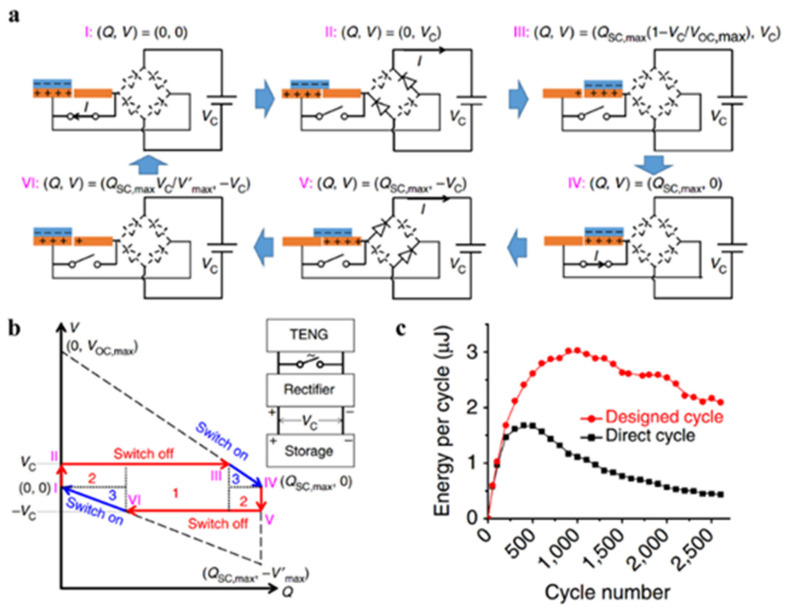
Power management with parallel travel switch. (**a**) Electrical connection and closing position of the parallel travel switch. (**b**) *V*-*Q* curve of the charging circuit with the switch. (**c**) The dependence of energy per cycle-on-cycle number with/without management. Reproduced with permission [70]. Copyright 2016, *Nature*.

**Figure 5 sensors-22-01668-f005:**
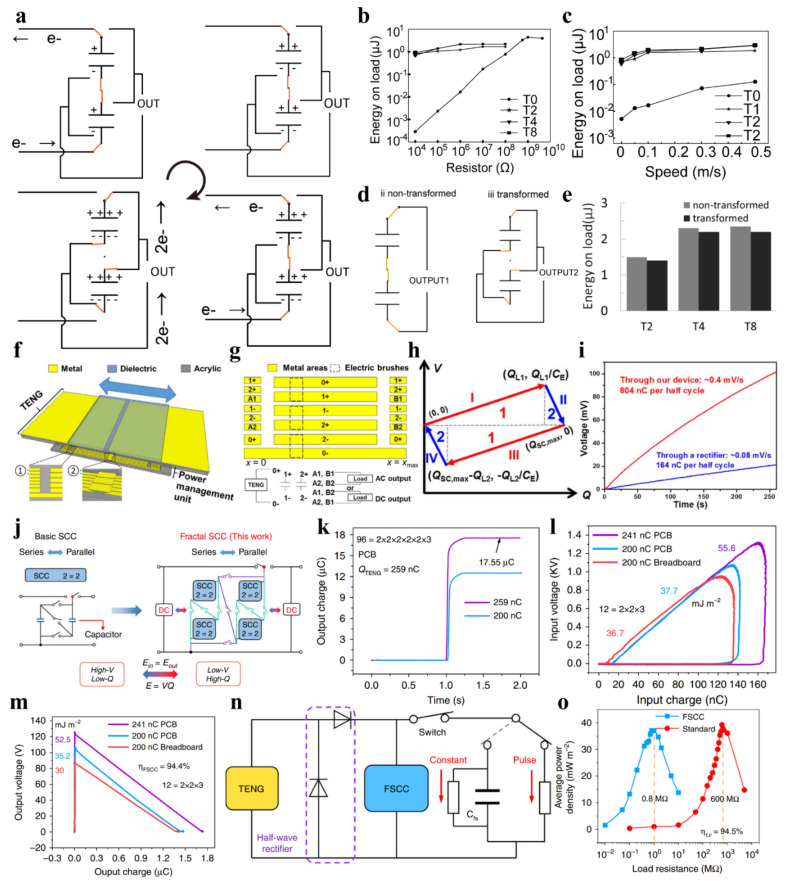
Switch capacitor convertor with travel switch. (**a**) The switching process and charge transfer of the first switch capacitor convertor for TENG. (**b**) Output energy on different load with/without power management. (**c**) Output energy on load at different speed with/without power management. (**d**) Electrical connection of transformed and non-transformed circuit. (**e**) Transformed and non-transformed energy output [60]. (**f**) Structure of the inductor-free triboelectric management method. (**g**) Structure of the travel switch. (**h**) *V*-*Q* curve of the managed energy output. (**i**) Comparison of the charged voltage between rectifier circuit and the power management circuit. Reproduced with permission [71]. Copyright 2017, Elsevier. (**j**) Fractal design. (**k**) Comparison of output charge between short-circuit and fractal design based switched-capacitor-convertors (FSCC) circuit. (**l**) Input *V*-*Q* curve and (**m**) output *V*-*Q* curve. (**n**) Electrical connection of constant mode and pulse mode with FSCC. (**o**) Comparison of dependence of average power density on load between FSCC and standard circuit. Reproduced with permission [64]. Copyright 2020, *Nature*.

**Figure 6 sensors-22-01668-f006:**
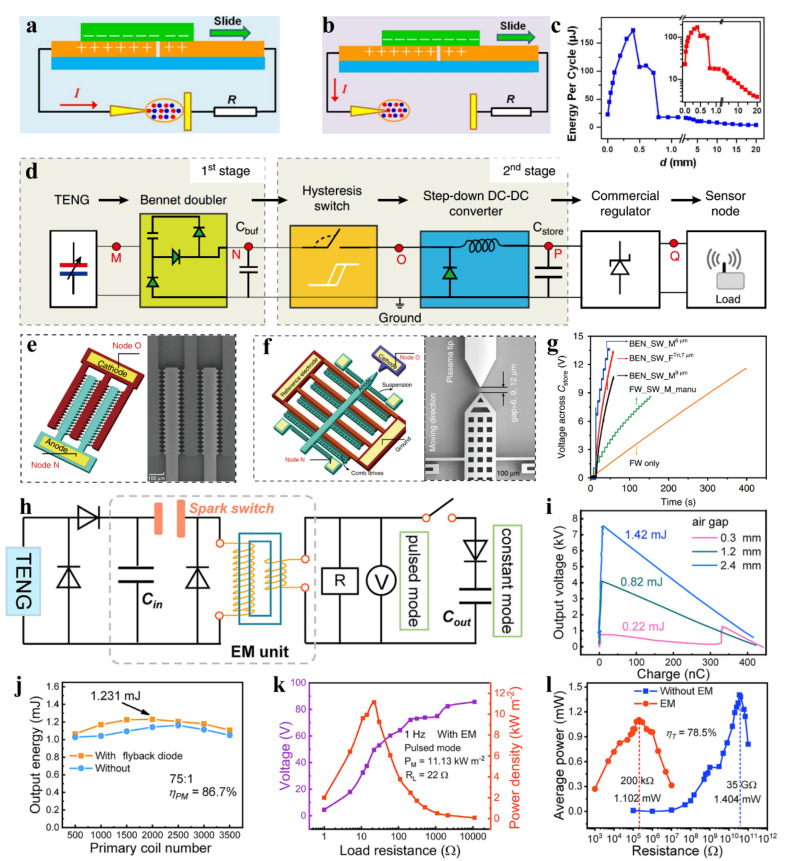
Power management with spark switch. (**a**) Arc discharge. (**b**) Corona discharge. (**c**) The relationship between the discharge energy per cycle and the electrode spacing. Reproduced with permission [72]. Copyright 2018, Elsevier. (**d**) The circuit architecture of the energy harvesting and conditioning. (**e**) Fixed switch. (**f**) Movable switch. (**g**) Comparison of charging rates for different switch configurations. Reproduced with permission [65]. Copyright 2020, *Nature*. (**h**) The circuit architecture of power management with spark switch. (**i**) Comparison of output energy by *C_in_* under different air gaps. (**j**) The dependence of output energy and transformer primary coil number. (**k**) The dependence of the voltage and power density on load resistance in pulse mode. (**l**) Comparison of average power between power management and standard circuit. Reproduced with permission [68]. Copyright 2021, Elsevier.

**Figure 7 sensors-22-01668-f007:**
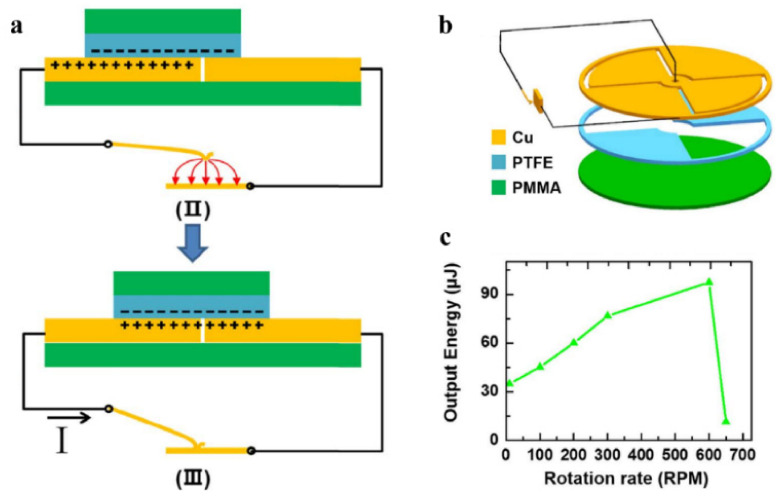
Power management with electrostatic switch. (**a**) Motion of cantilever vibration switch under electrostatic attraction. (**b**) Turntable TENG with the electrostatic switch. (**c**) The dependence of output energy on rotation rate. Reproduced with permission [73]. Copyright 2018, Elsevier.

**Figure 8 sensors-22-01668-f008:**
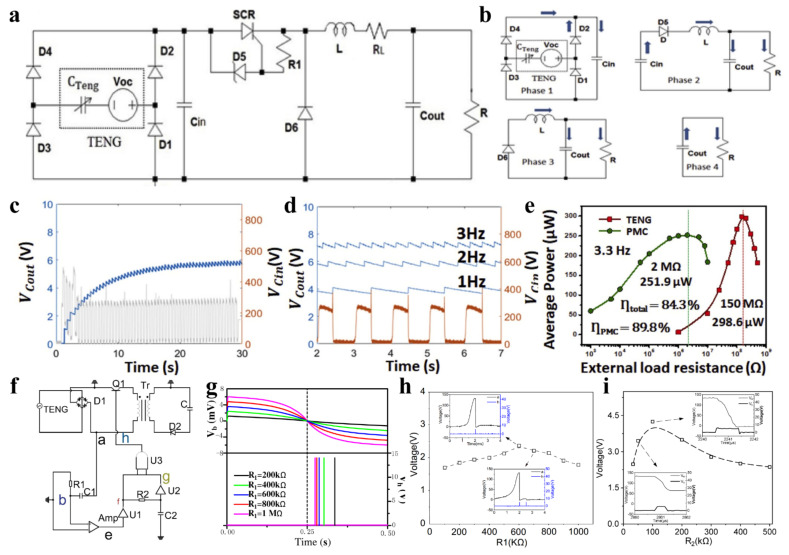
Power management with silicon-controlled rectifier (SCR) and triode. (**a**) Power management circuit with SCR. (**b**) Energy flow in four stages. (**c**) The voltage waveforms of *C_in_* and *C_out_*. (**d**) The dependence of *C_in_* and *C_out_* voltage on frequency. (**e**) Average output power under different loads with/without power management. Reproduced with permission [66]. Copyright 2020, Elsevier. (**f**) Power management circuit with triode. (**g**) The dependence of *V_b_* and *V_h_* on different value of *R*_1_. (**h**) The optimum value of *R*_1_. (**i**) The optimum value of *R*_2_ [74].

**Figure 9 sensors-22-01668-f009:**
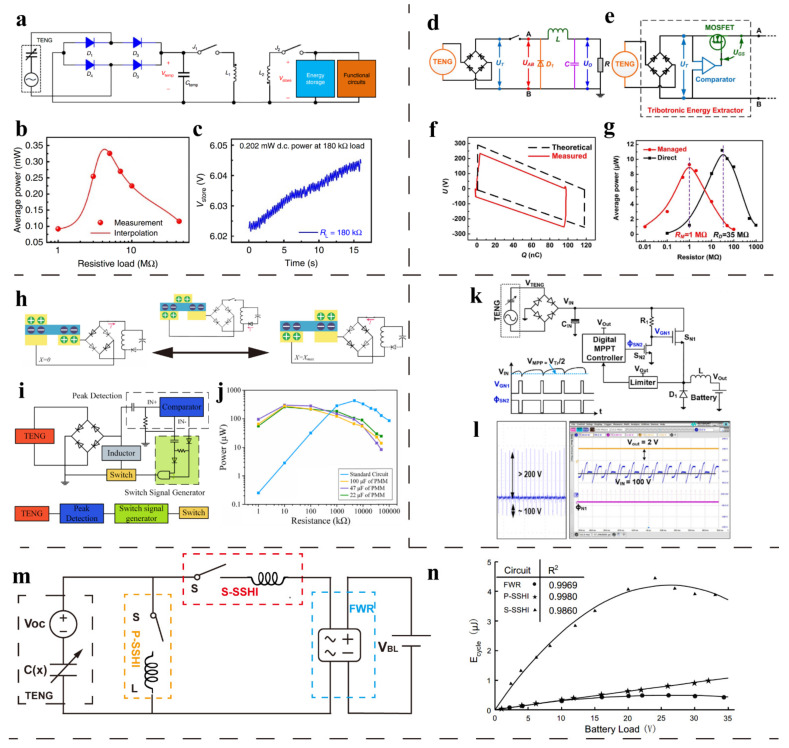
Power management with MOSFET switch. (**a**) The two-stage energy releasing power management circuit. (**b**) Relationship between average power and resistance before management. (**c**) The optimum DC power is obtained at 180 kΩ load. Reproduced with permission [62]. Copyright 2015, *Nature*. (**d**) Power management circuit combining TENG maximized energy output cycle (CMEO) and buck convertor. (**e**) Basic configuration of the tribotronic energy extractor. (**f**) Theoretical and measured *V*-*Q* curve. (**g**) The output matching impedance of TENG before and after the power management. Reproduced with permission [63]. Copyright 2017, Elsevier. (**h**) The circuit of power management method and working process of the switch. (**i**) Basic circuit topology. (**j**) Output impedance-power diagram under different PMMs and standard circuit. Reproduced with permission [76]. Copyright 2019, Elsevier. (**k**) A triboelectric power management circuit based on maximum power point tracking (MPPT). (**l**) Output waveforms of TENG open-circuit voltage, battery voltage, input capacitor voltage and switching control signal. Reproduced with permission [77]. Copyright 2021, Elsevier. (**m**) Circuit topology of P-SSHI, S-SSHI and FWR. (**n**) The dependence of energy per cycle on voltage of battery load [78].

**Figure 10 sensors-22-01668-f010:**
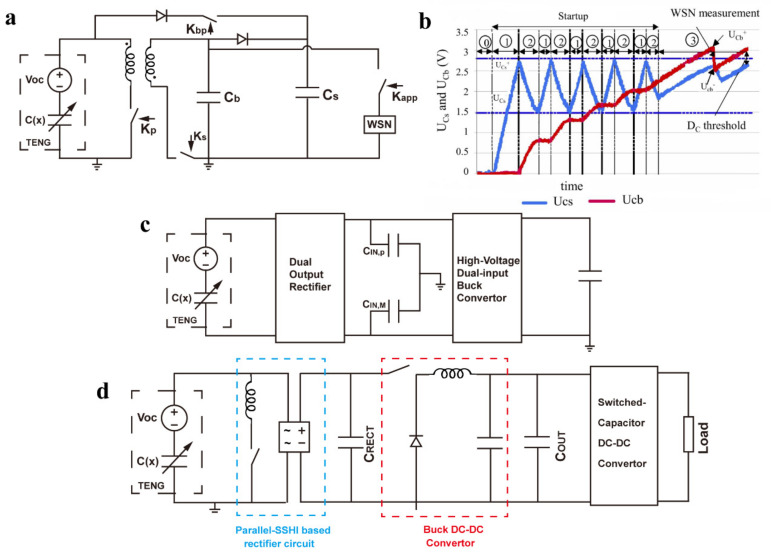
Power management with integrated circuit. (**a**) A self-starting power management integrated circuit for triboelectric energy harvesting. (**b**) Voltage waveform of capacitor *C_b_* and *C_s_* [79]. (**c**) A high-voltage dual-input integrated circuit converter for power management [80]. (**d**) A triboelectric energy harvesting circuit based on synchronous inductor parallel switch and DC step-down conversion [81].

## Data Availability

Not applicable.

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
