# Peer review of "Recent Progress of Switching Power Management for Triboelectric Nanogenerators"

_sensors, 2022, doi:10.3390/s22041668_

Round 1
Reviewer 1 Report
This manuscript authored by H. Zhou et al systematically introduced the recent progress of switching power management for triboelectric nanogenerator(TENG). The review of switching power management for TENGs focused on the switch structure, power management methods of four categories, and advantages and drawbacks of various switching power management circuits for TENGs. Overall, this work is interesting, and the authors have provided a systematic study on the topic without critical defects. In addition, review for the design ideas and power management results of the reported works provides prospects for the future development of TENG power management and TENGs for various applications. Therefore, reviewer recommends this manuscript is acceptable to sensors minor revision process as below comments.
Comment 1: “The authors describe theoretical guidance for TENG power management in manuscript to aid the reader's understanding. However, the current introduction lacks in detailed foundational principles about TENG to understand the theoretical guidance for TENG. In order to convey more useful information to the reader, the author should describe to the mechanism and working principle of TENG in the introduction part.”
Comment 2: “The size of the graph in the figure 1, 4, and 8 is too small. I suggest that size of figures should be enlarged to catch reader’s eyes.”
Comment 3: “3. I suggest that authors should compare advantages and drawbacks of switching power management circuits for TENGs with dividing into mechanical and electronic switches in the conclusion part.”
Reviewer 2 Report
The paper is well written and can be accepted for publication with minor changes.
1. The authors should add a graphical illustration of the progress in the power management circuit over years and also discuss the disadvantage of each system and how to overcome them as future prospects.
2. They are recommended to cite two recent works in triboelectric generators such as Nano Energy 91, 106662, 2021 and Nano Energy 88, 106255, 2021.
